# Refining Fire–Climate Relationship Methodologies: Southern California

**Benjamin Bleiman [1],\*, Tom Rolinski [2], Eric Hoffman [1], Eric Kelsey [1]**  **and David Bangor [2]**

[1] Department of Meteorology, Plymouth State University, Plymouth, NH 03264, USA; ehoffman@plymouth.edu (E.H.); ekelsey2@plymouth.edu (E.K.)

[2] Southern California Edison, Rosemead, CA 91770, USA; tom.rolinski@sce.com (T.R.); david.i.bangor@sce.com (D.B.)

\* Correspondence: bobleiman@plymouth.edu

**Abstract:** Efforts to delineate the influence of atmospheric variability on regional wildfire activity have previously been complicated by the stochastic occurrence of ignition and large fire events, particularly for Southern California, where anthropogenic modulation of the fire regime is extensive. Traditional metrics of wildfire activity inherently contain this stochasticity, likely weakening regional fire–climate relationships. To resolve this complication, we first develop a new method of quantifying regional wildfire activity that aims to more clearly capture the atmospheric fire regime component by aggregating four metrics of fire activity into an annual index value, the Annual Fire Severity Index (AFSI), for the 27-year period of 1992–2018. We then decompose the AFSI into trend and oscillatory components using singular spectrum analysis (SSA) and relate each component to a set of five climate predictors known to modulate macroscale fire activity in Southern California. These include the Atlantic Multidecadal Oscillation (AMO), Pacific Decadal Oscillation (PDO), El Niño–Southern Oscillation (ENSO), and Santa Ana wind (SAW) events, and marine layer frequency. The results indicate that SSA effectively isolates the individual influence of each predictor on AFSI quantified by generally moderate fire–climate correlations, $|r| > 0.4$, over the full study period, and $|r| > 0.5$ over select 13–15-year periods. A transition between weaker and stronger fire–climate relationships for each of the oscillatory PC–predictor pairs is centered around the mid-2000s, suggesting a significant shift in fire–climate variability at this time. Our approach of aggregating and decomposing a fire activity index yields a straightforward methodology to identify the individual influence of climatic predictors on macroscale fire activity even in fire regimes heavily modified by anthropogenic influence.

**Keywords:** wildfire; fire–climate relationships; climate; weather; Southern California; human influence; singular spectrum analysis

## 1. Introduction

In Southern California, where it is estimated that ~95% of all wildfire ignitions are human caused [1], trends in fire activity are often decoupled from the climate and weather patterns that shape the regional fire regime. The stochastic nature of ignition and large fire events complicate efforts to elucidate these "fire–climate relationships" [2]. Non-climatic factors, including resource availability, suppression strategy, and incident prioritization, can influence the size of individual fires, particularly when suppression resources are overwhelmed in periods of widespread fire activity [3]. Therefore, the metrics typically used to quantify fire activity, including the number of ignitions and the number of acres burned may not yield clear relationships with regional climate drivers. To improve model forecasts of regional fire activity, supplementary metrics are needed that correlate more strongly to regionally important climate patterns.

Fire activity can be understood fundamentally as enabled and driven by climate and weather, respectively [3]. Climate patterns have a strong yet complex influence on fire

occurrence and extent because they control fuel production, composition, and moisture [4,5]. Southern California's Mediterranean climate consists of strong seasonal variations in precipitation and temperature wherein the large majority of annual precipitation occurs in the cool season (roughly October–March), regularly leading to 6 or more months of very dry conditions during the warm season (April–September). Abundant cool season precipitation enhances fine fuel production, whereas the advance of the following warm season increases the available fuel load due to continually declining fuel moisture and increasing fuel flammability [6].

These annual climate patterns are modulated by several synoptic-scale ocean/atmosphere teleconnections that are known to modify the fuel production and moisture cycles of Southern California. Sea surface temperature (SST) anomalies in both the Atlantic and Pacific basins drive atmospheric responses that influence patterns of precipitation, drought, and wildfire activity in Southern California. The Atlantic Multidecadal Oscillation (AMO) is an index that represents one of these modes of Southern California climatic variability in the northern Atlantic Ocean [7,8], whereas the Pacific Decadal Oscillation (PDO) and El Niño–Southern Oscillation (ENSO) represent these modes of regional climate variability in the Pacific Ocean [5–7,9–14]. Warm/positive AMO phases are linked to widespread drought and synchronous fire activity across the Western US, including Southern California, whereas cool/negative PDO and ENSO are linked to increased drought and wildfire tendencies in Southern California (though the physical mechanism by which these respective oscillations influence climate patterns differ). The respective oscillations further influence regional fire activity via the constructive or destructive influence of the attendant atmospheric responses from each oscillation [6,7,9,10,13]. For example, extended droughts have been found during concurrent positive AMO, negative PDO, and negative ENSO phases, leading to extended periods of abnormally low fuel moistures [7,8], and thus favorable conditions for widespread fire activity.

At shorter timescales, two prominent weather patterns are critical in modulating fuel moisture and thus fire activity in Southern California: first, the hot, dry foehn winds known locally as Santa Ana winds (SAWs), which occur frequently in the cool season and can drive extreme fire weather conditions [11,15–17]; and second, the frequent warm season occurrence of coastal stratus clouds that form within the cool, humid marine boundary layer [18]. Though the influence of these phenomena is predominantly confined to a narrow area between the coast and nearby mountain ranges, these areas contain much of the regional fire activity due to the abundance of available fuels, expansive wildland–urban interface (WUI), and abundant human-caused ignition potential [1,17,19,20].

The inherently unpredictable nature of when and where anthropogenic ignitions occur is one of the primary drivers of the Southern California fire regime [1,21]. Though weather- and climate-scale drought regularly desiccate fuels, fire impacts as they are traditionally defined (ignitions and acres burned) are not realized unless a stochastic ignition occurs. More specifically, these traditional metrics quantify only physical impacts that are *observed*, without accounting for *potential* physical impacts. Hypothetically, by representing annual fire severity as an aggregation of multiple indices, each of which quantify both observed and potential fire activity, the stochastic nature of ignitions and large fire events will not dominate fire–climate relationships.

This research first introduces our Annual Fire Severity Index (AFSI), a novel method of quantifying observed annual fire activity that aggregates into a single index value the two standard indices of severity (ignitions, acres burned) plus two measures of the potential for fire suppression resources to be strained, which we believe are captured by the magnitude of the energy release component (ERC) and the fire season length. ERC is an output of the National Fire Danger Rating System (NFDRS) [22,23] employed by fire managers countrywide that quantifies potential heat release at a fire's head by considering the cumulative drying effect of weather in the previous days to weeks through the incorporation of live and dead fuel moisture. ERC is closely related to large fire occurrence and extreme fire behavior [2]. Fire season length quantifies the time period

wherein regional fuel conditions are conducive for significant wildfire activity and thus the potential strain on fire suppression resources given the occurrence of an ignition. A longer fire season is more likely to strain financial resources, fire suppression crews, and the overall effectiveness of fire containment.

The second goal of this research is to develop a methodology for correlating this AFSI with a set of atmospheric predictors that serve to succinctly, but comprehensively, represent the major atmospheric elements of the fire regime. We hypothesize that the annual fire activity time series can be modeled as the sum of individual time series that each isolate the influence of one climatic feature on fire activity. This allows us to identify components of the fire regime that oscillate at varying frequencies and magnitudes, incorporating indices of weather- and climate-scale phenomena collectively. To do so, we employ a time-series decomposition technique, singular spectrum analysis (SSA). Our novel approach in re-defining and quantifying regional fire–climate relationships serves to advance modeling of macroscale fire–climate relationships.

## 2. Study Area

This analysis focuses on the Geographic Area (GA) boundaries defined for Southern California (Figure 1), referred to herein as the Southern California Geographic Area (SCGA) for simplicity. GA boundaries are designated by intergovernmental wildland fire protection agencies for "planning, coordination, and operations leadership for effective utilization of emergency management resources within their area [24]." There are ten such GAs throughout the United States. GA boundary shapefile data were collected from the National Wildfire Coordinating Group (NWCG).

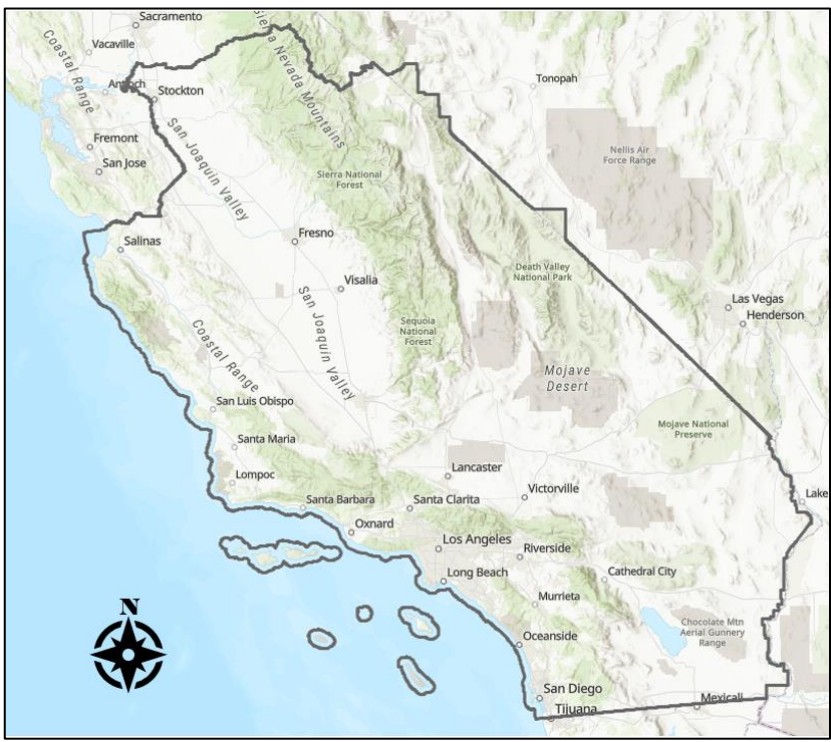

**Figure 1.** The Southern California Geographic Area (SCGA), with notable geographic features labeled.

We justify focusing our analysis on the GA scale for two primary reasons. First, understanding fire–climate relationships at the spatial level of a GA corresponds to the scales at which climate information is digested and interpreted by fire management operations for seasonal planning, resource allocation, and suppression efforts [25,26]. Second, correlations of fire–climate relationships have generally been found to increase with a growing area of study. It has been hypothesized that the variability of fire and climate data becomes attenuated as the level of aggregation increases, thus allowing the fire–climate relationships

to emerge more clearly [3,27]. This methodology allows for another layer of aggregation from which a clearer understanding of macroscale fire–climate relationships can be derived. The term "macroscale" is used here to differentiate from the small scale, on-the-ground influences of factors including heat and moisture fluxes on fire activity that are not the focus of this research. Our research focuses on macroscale effects of climate patterns on fire activity across broad fire regimes, relationships which are much more readily established at large geographic scales.

## 3. Data and Methods

### 3.1. Development of Annual Fire Severity Index

#### 3.1.1. Wildfire Data

The first two components of the AFSI, wildfire ignition counts and acres burned, were derived from the Fire Program Analysis fire-occurrence database (FPA FOD), 5th edition [28]. The FPA FOD includes all fires across the United States from 1992 to 2018 and integrates fire records from federal, state, and local fire organizations. Two filters were applied to the dataset: First, all fires < 1 acre were excluded to remove noise from the overall signal of fire activity. Second, fires ignited on days with low fire energy intensity, represented by $ERC < 50$, e.g., [29] were also excluded. This threshold eliminates roughly the bottom 5% of all fires $\geq 1$ acre. For all fires $\geq 1$ acre ignited when daily $ERC < 50$, only about 3.6% and 0.5% exceeded final sizes of 100 and 1000 acres, respectively, and thus are very unlikely to contribute significantly to annual fire activity. The final dataset included roughly 42,000 fires.

#### 3.1.2. ERC Data

The remaining two AFSI components, yearly ERC magnitude and fire season length, were both calculated from a daily ERC time series gathered from the web-based cloud-computing application Climate Engine [30]. A time series of daily ERC values spatially averaged over the SCGA was downloaded. These ERC data were derived from the University of Idaho's Gridded Surface Meteorological Dataset (gridMET, [31]). Although the ERC magnitude component and the fire season length component were derived from the same ERC time series, these two metrics are not strongly correlated and were thus both included in the AFSI calculation.

#### 3.1.3. AFSI Calculation Methodology

Our AFSI comprises annual values of each of the four components, two derived from the FPA FOD (acres burned, ignition counts) and two derived from the ERC database (ERC magnitude, fire season length). These four components were preliminarily selected based on their perceived ability to quantify different but equally important modes of annual wildfire severity related to observed fire activity (which contain an element of anthropogenically-derived stochasticity) and potential fire activity (which are related only to atmospheric conditions and the attendant impacts on fuel flammability). Each of the four metrics were established throughout the fire science community and can be easily attained and interpreted.

An exploratory analysis was conducted on each of the components' empirical distributions to identify their most appropriate quantification, keeping in mind that we aim to comprehensively represent the different modes of annual fire activity and their potential impact on fire suppression efforts. The first component, acres burned, was calculated as the annual number of fires exceeding the climatological (1992–2018) 97th percentile fire size. This focuses on the annual fire activity contribution by very large fires, which can require a disproportionate suppression resource response compared to smaller fires. The second component, annual ignition count, was left unaltered because it directly represents the potential for many concurrent fires to strain fire suppression resources. Third, we determined that the ERC magnitude component was best represented by the yearly 90th percentile ERC value. Daily ERC values, which ranged from about 10 to 95 throughout the

study period, displayed a right-skewed distribution, indicating that high daily ERC values were more frequent. The 90th percentile value was less prone to sampling variation than a lower percentile, such as the median, because of this right skew.

The calculation of the fourth AFSI component, fire season length, was necessarily more subjective due to the complex nature of ERC time series and the fuel conditions they represent. Highly variable winter rainfall interspersed with frequent SAW events across Southern California commonly result in fuel moisture values that fluctuate greatly from day to day in the cool season. We chose to emphasize the assumed strain on suppression resources of many consecutive days with elevated fire potential. The daily ERC time series was smoothed with a Savitzky–Golay (SG, [32]) filter of window length $l = 49$ days and polynomial order $p = 2$ (Figure 2). This filter is advantageous in its ability to preserve narrow time-series peaks better than many other smoothing filters with Gaussian convolution [33]. We then calculated the annual number of consecutive days where $ERC \geq 50$, an empirical threshold under which we determined that significant fire activity was unlikely. However, it is common for multi-day periods to exceed this empirical threshold outside of the primary fire season; therefore, these sub-periods were included in the respective year's fire season count if they remained within 15 days of the beginning or end of the primary fire season. Fire season tallies were cut off on 31 December of each year.

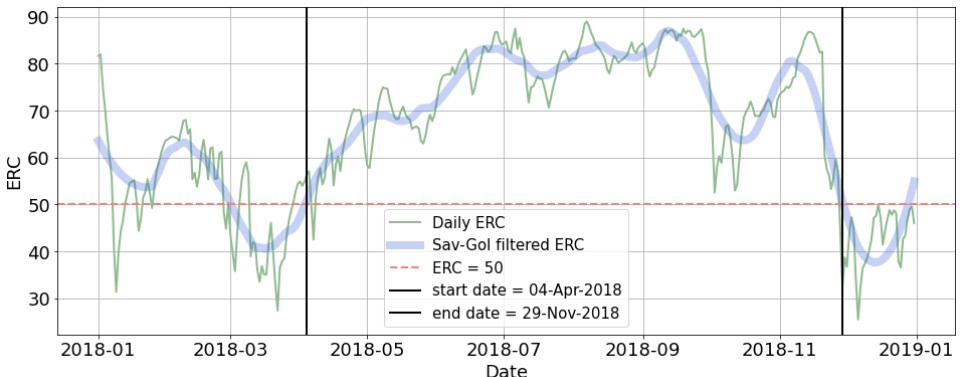

**Figure 2.** Energy Release Component (ERC) time series for calendar year 2018 over the Southern California Geographic Area (SCGA), including daily values (thin green line), Savitzky–Golay filtered values (ERC$_{SG}$; thick blue line), and the beginning and end of the "fire season," as determined by the consecutive period where $ERC_{SG} \geq 50$.

In considering how each of the AFSI components operate together, it is useful to note that the respective variance of each component is maximized during a different portion of the calendar year. This characteristic allows each component to represent intra-annual modes of fire activity that may occur in the absence of another and therefore may be missed when considering only ignitions and acres burned. For example, the monthly distribution of ERC magnitude compared to ignition frequency shows that both components were maximized in the summer when SCGA weather conditions were consistently hot and dry (Figure 3). Yet within these summer months, interannual ignition frequencies varied widely, whereas ERC values were consistently elevated; oppositely, the winter months exhibited a consistently low ignition frequency, whereas ERC magnitude varied drastically because of the complex interference between variable winter rainfall and SAW events. Each AFSI component is hypothetically capable of capturing very different behaviors that can each lead to significant fire activity given a stochastic ignition.

Once the AFSI components were selected, the index time series was calculated using the following method. Each of the four components were assigned a yearly "score" from 1 to 10, determined by the yearly components' decile value with respect to its 27-year climatology. A yearly score of 1 was assigned if the yearly component value fell within the bottom decile of the component distribution, whereas a yearly score of 10 was assigned if the yearly component value fell within the top decile of the component distribution,

and so on. This calculation was applied to each of the four AFSI components and the yearly component values were summed. The result was a time series of 27 annual values ("scores") ranging from 4 to 40, where a score of 4 indicates the least severe fire season possible and a score of 40 indicates an exceptionally severe fire season.

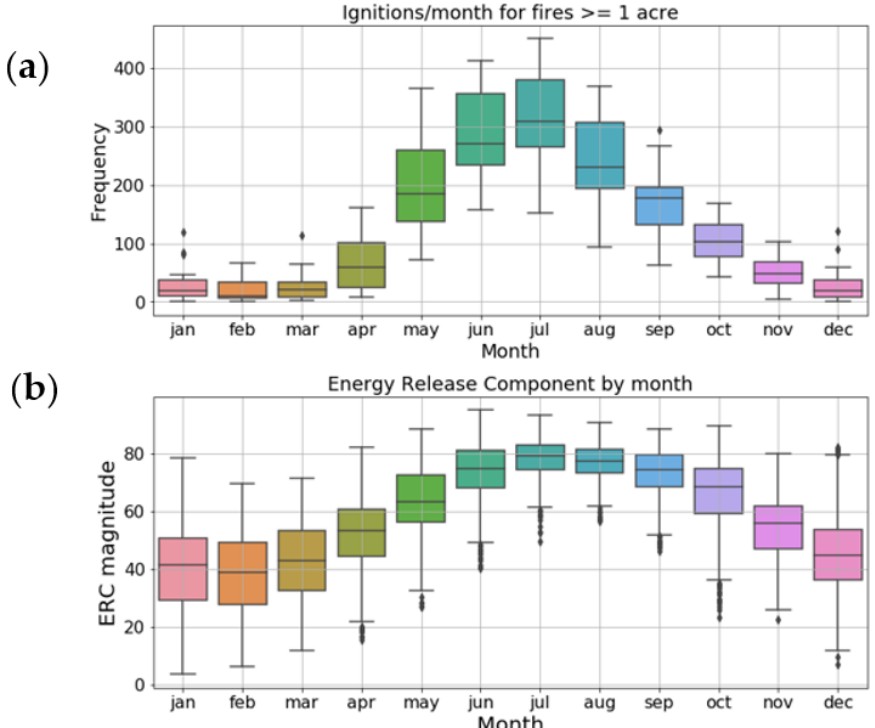

**Figure 3.** Box-and-whisker plots of SCGA monthly (**a**) fire ignitions and (**b**) ERC magnitude, two of the components used in the Annual Fire Severity Index (AFSI) calculation, for the study period 1992–2018. Boxes indicate the 25th, 50th, and 75th percentiles of the distribution, whereas the whiskers extend to the 5th and 95th percentiles. Black dots indicate outlier values.

The final AFSI time series for the 27-year period, 1992–2018, is displayed in Figure 4, and a diagram depicting the AFSI development workflow is displayed in Figure 5. Examining the AFSI time series, the range in annual score sum spanned from a minimum value of 7 in 1998 to a maximum of 37 in 2007. The index score distribution was roughly normal with *median* = 22 and *mean* = 21.85. Second, the time series could be divided into separate periods of generally elevated or suppressed AFSI values. From 1992–2004, scores generally remained at or below the median, particularly from 1998 to 2001, where three of four years were well below the median. A period of elevated scores then prevailed from 2005 to 2018, beginning with three years of scores increasing 8, 8, and 5 points, respectively, from 2005 to 2007.

If the impact of anthropogenic activity (human ignitions, fire suppression resource management/placement, etc.) on the AFSI had been effectively diluted, we would expect that the annual scores and overall time series behavior were largely driven by SCGA weather and climate features. Hypothetically, we can presume that (1) the AFSI time series represents the purely climatic signal of SCGA fire activity; (2) that we can decompose the time series into components representing trend, oscillatory behavior, and noise; and (3) that these time-series components can be related to the set of weather and climate patterns we have identified as significant in modulating SCGA fire activity. The following section details the statistical analysis methods used to relate climate predictors to the AFSI component predictands.

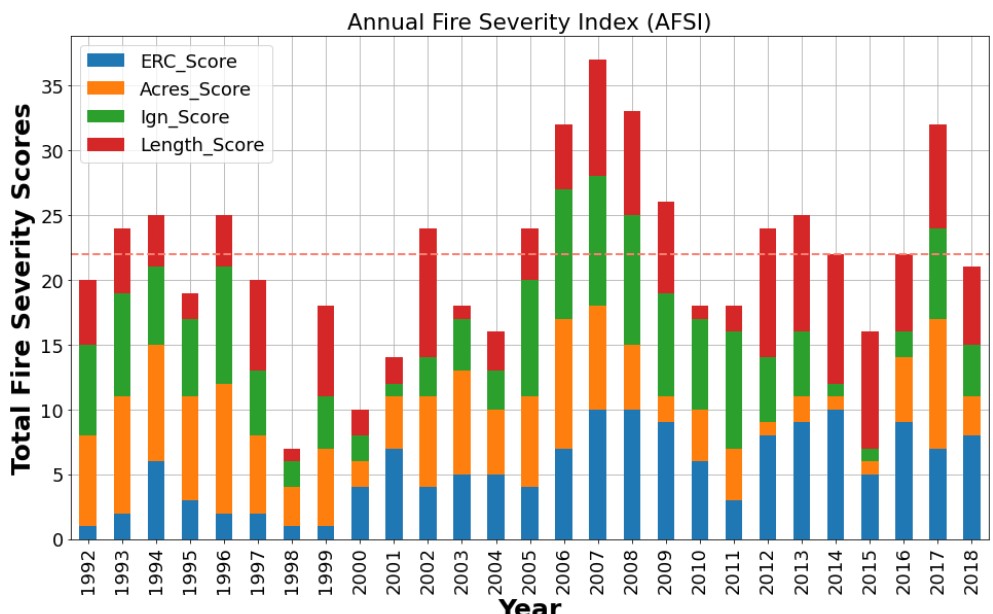

**Figure 4.** The calculated AFSI time series for study period 1992–2018. Each of the four AFSI components are indicated by different colors (ERC, blue; acres burned, orange; ignitions count, green; and fire season length, red). Yearly component scores are stacked to indicate the yearly sum score. The 27-year mean value, 21.85, is displayed as the horizontal dashed line.

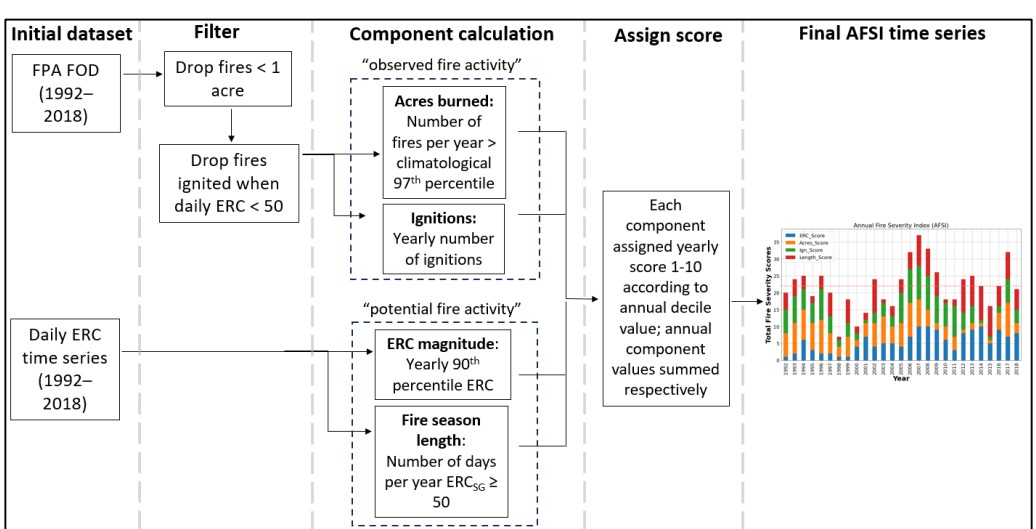

**Figure 5.** A schematic depicting the 5-step AFSI development workflow from the initial dataset to the final time series as described throughout Section 3.1. First, the Fire Program Analysis fire occurrence database (FPA FOD) dataset is collected and two filters are applied. The acres burned and ignitions components are then calculated as indicated and collectively serve to represent "observed" fire activity. Next, the unfiltered daily ERC time series is used to calculate the ERC magnitude and fire season length components as indicated, where $ERC_{SG}$ represents the Savitzky–Golay filtered ERC values (Figure 2). These two components represent "potential" fire activity. Then, each annual component value is assigned a score from 1 to 10, the respective component values are summed, and the final AFSI time series is plotted.

### 3.2. Statistical Analysis of AFSI Fire–Climate Relationships

3.2.1. Atmospheric Predictor Data

A set of five atmospheric predictors was selected as collectively responsible for a large portion of the interannual variability in SCGA fire activity, encompassing the primary

drivers of large-scale fuel moisture content in space and/or time. Comprising this set of atmospheric predictors are three modes of large-scale climatic variability in the AMO, PDO, and ENSO ocean–atmosphere teleconnections, plus two weather phenomena that are less widespread in spatial influence but important in modulating fuel moistures within the most fire-prone regions of the SCGA, represented by marine layer and SAW frequency.

AMO Index data (unsmoothed, detrended) for the period 1948–2022 were downloaded from the National Oceanic and Atmospheric Administration (NOAA) Physical Sciences Laboratory (PSL; [34]). This index is calculated from the Kaplan SST dataset, Version 2, and represents SST anomalies in the North Atlantic. PDO Index data were downloaded from the National Centers for Environmental Information (NCEI) and are based on the NOAA's extended reconstruction of the Extended Reconstruction SST (ERSST Version 5) and represent SST anomalies in the northeast and tropical Pacific Ocean [35]. The SG filter was again applied to the AMO and PDO time series with window length $l = 61$ months and polynomial order $p = 3$. The window lengths were chosen to reduce short-term variability while maintaining the integrity of the overall phases and phase transitions. Annual AMO and PDO values were then calculated as the median monthly value of each year. ENSO conditions were quantified herein by the Oceanic Niño Index (ONI), the NOAA's primary indicator for the monitoring of ENSO conditions, and were collected from the NOAA Climate Prediction Center's (CPC) ONI database [36]. This index represents the 3-month running average of ERSST Version 5 in the Niño 3.4 region. ENSO teleconnections with weather/climate in the Western US are typically strongest in the winter months [37], and so December–February mean ONI values were calculated to represent the annual ONI state, e.g., [10].

It is important to quantify SAWs and their ability to rapidly dry fuels and drive extreme fire activity separately from climate timescale moisture anomalies, which are related to atmospheric forcing by AMO, PDO, and ENSO. We believe that the most representative method to quantify high-frequency variations in fire potential due to fuel drying, such as those related to SAW events, is the Evaporative Demand Drought Index (EDDI; [38,39]). EDDI considers only evaporative demand ($E_0$) anomalies with respect to the climatological median over a range of time windows from 1 week to 12 months. At the shorter end of the spectrum, EDDI at 2-week and 1-month time scales closely corresponds to documented heat waves and extreme fire weather in California [39] and therefore can serve as a proxy for SAW events that are likely to rapidly reduce fuel moistures and increase fire potential. Following these observations, our EDDI parameter was calculated as the number of times per year where the 2-week $EDDI > 1$ as averaged over SCGA. Although we also tested an annual SAW occurrence time series [15] to identify the direct influence of SAW frequency on fire activity, fire–climate correlations were generally weaker than those with our EDDI time series.

Finally, we calculated an index of annual marine layer frequency over the study period to determine relationships with fire activity, following the methodology employed by [18]. However, no significant correlations were detected.

### 3.2.2. Singular Spectrum Analysis

To decompose the AFSI time series, we employed singular spectrum analysis (SSA; [40–44]), a technique that has been used extensively on geophysical and climate data. SSA decomposes a time series into its underlying components (trend, oscillatory modes, noise) and is especially useful for decomposition of short, noisy time series. As a non-parametric spectral estimation method, SSA makes no assumptions about the parametric form of the time series. SSA uses data-adaptive basis functions and therefore can separate potentially irregular oscillatory behaviors with varying periodicities [45]. This is advantageous because the atmospheric features that control SCGA weather and climate vary irregularly and may not be clearly identified by decomposition methods that assume sinusoidal behavior. SSA is essentially principal component analysis (PCA) on a time series, identifying the primary modes of temporal variability from within the data. The main result is a set of orthogonal

reconstructed components (RCs), herein referred to as principal components (PCs), which together sum to the original time series.

SSA is also practical and easy to employ because it requires only one variable, the "window length," or the number of consecutive time series observations $n$ upon which a delay window is applied, and determines the longest periodicity captured by SSA. Given a window length of $L$, SSA produces $L$ principal components. Larger values of $L$ provide more refined decomposition into elementary components and therefore better separability [46]. To sufficiently extract the underlying trend of the AFSI, a window length of $L = 12$ was chosen. Therefore, the window length consisted of about 44% of the overall study period length. The SSA code implemented herein was adapted from a Python script by D'Arcy [47].

We hypothesize that it is possible to isolate the forcing signals of the atmospheric predictors from within the AFSI using SSA. By examining the periodicities and relative contributions of each SSA component to the overall time series and relating these behaviors to those of a weather/climate predictor, it may be feasible to identify a physical PC–predictor connection. In presenting the AFSI statistical analysis results, the discussion begins with the SSA decomposition and the apparent behaviors of each PC. Then, each PC is identified by its relationship to the set of weather and climate predictors. Time series of each predictor and PC are first plotted to observe patterns within the data, and the relationships are then quantified using the Pearson correlation coefficient ($r$). A thorough examination of potential lagged fire–climate relationships, contingent relationships of weather/climate features with each other, and other complications that may have weakened the results follow.

## 4. Results

### 4.1. SSA Decomposition

The SSA decomposition of the 27-year AFSI time series using a window length of $L = 12$ produced 12 PCs, referred to herein as Components 0–11 (Comp0–11). Though some signal may have been contained in Comp7–11, their respective variance contribution was very small and showed insignificant correlations with our predictors. Accordingly, Comp7–11 were excluded from this analysis. Cross-component correlations guided our subjective determination to group Comp1 with Comp2 (Comp12), Comp3 with Comp4 (Comp34), and Comp5 with Comp 6 (Comp56). Figure 6 displays the resulting PCs over the study period and Table 1 provides statistics of the grouped temporal PCs. Comp0 exhibited a strongly linear behavior throughout the study period, whereas each of the remaining components exhibited oscillatory behavior largely centered around mean = 0. Therefore, SSA effectively isolated the trend and oscillatory components with our choice of $L$.

Given the set of weather/climate predictors (AMO, PDO, ENSO, SAW day counts, and marine layer frequency), our aim was to identify each of the PCs as largely driven by an individual predictor. This could be accomplished by examining the PC behaviors, specifically their magnitude, periodicity, and contribution to the AFSI as compared to observed contributions of the weather/climate parameters toward fire activity within the literature.

#### 4.1.1. Component 0

Comp0 exhibited a positive trend throughout the study period, rising from a minimum value of 17 points in 1992 to a maximum value of 27 in 2017 before slightly decreasing in the final year, 2018 (Figure 6). Comp0 represented a very large portion of the overall time-series variance at 91.15% (Table 1), likely indicating that it is driven by a fire–climate component with wide-ranging spatial and temporal influence on the SCGA fire regime.

Comp0 was identified here as strongly correlated to AMO ($r = 0.741$, $p < 0.001$). In a negative (cool) phase through about 1996, AMO then transitioned to a positive (warm) phase from 1996 through the end of the study period in 2018 (Figure 7). Examining the full AMO period of record revealed that since reaching its strongest negative phase in the mid-1970s, index values increased steadily into the 21st century (not shown). Thus, the

clear positive trend in AFSI values as isolated by Comp0 coincides with the AMO transition from a prolonged cool to warm phase. AMO's warm phase is known to synchronize fire activity at multidecadal time scales across western North America [7], a pattern seemingly reflected in the AFSI decomposition.

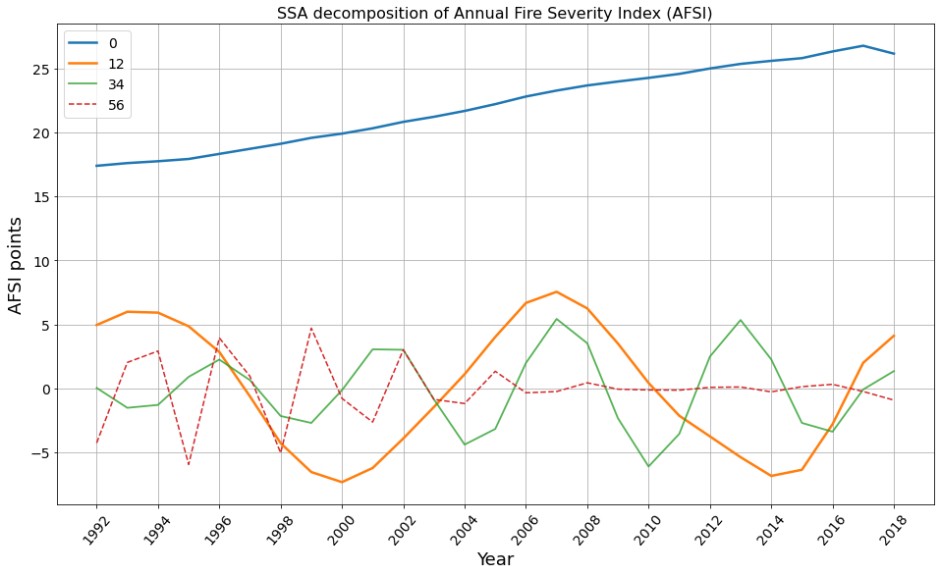

**Figure 6.** Each of the 4 grouped temporal Principal Components (PCs; Comp0, Comp12, Comp34, and Comp56) from the SSA decomposition of AFSI, 1992–2018.

**Table 1.** Statistics of the 4 grouped temporal PCs from the singular spectrum analysis (SSA) decomposition of the AFSI and their respective temporal characteristics (i.e., linearity vs. periodicity), overall AFSI variance contribution, and AFSI non-trend variance (excluding Comp0) to emphasize the relative contribution of the remaining SSA components.

|  | Principal Components | Characteristic | Variance (%) | Non-Trend Variance (%) | Total Variance (%) |
|---|---|---|---|---|---|
| *AFSI* | 0 | Trend—linear | 91.15 | - | 99.71 |
|  | 1–2 | Periodic (14–15 years) | 4.91 | 55.48 |  |
|  | 3–4 | Periodic (5–6 years) | 2.39 | 27.01 |  |
|  | 5–6 | Irregular periodic (2–3 years) | 0.98 | 11.07 |  |

Interestingly, the correlation between AMO and Comp0 further increased when considering a multi-year lag. This correlation was maximized at $r = 0.856$ ($p < 0.001$) with a 10-year lag, where AMO values from 1982 to 2008 and Comp0 values from 1992 to 2018 were related (Figure 8). Although multi-year lags between oscillation phase shifts and weather/climate teleconnections are documented in the literature, e.g., [48,49]), explicit support for lagged AMO effects up to 10 years on the Western US climate was not abundant as best as we can tell. However, given the multi-decadal periodicity of AMO phases and the large-scale influence of phase shifts on hemispheric teleconnections [50], decadal lag between AMO and fire activity seems plausible.

Although Comp0 was responsible for 91.15% of the overall reconstructed time-series variance, it must be stressed that we do not believe that AMO, in turn, directly contributed to annual fire severity so drastically. Other research which investigated AMO in relation to drought and/or fire activity across western North America have found AMO to be responsible for roughly 25–40% of variance [8,51–53]. It remains plausible that AMO greatly influenced SCGA fire activity over this 27-year study period because of its large-scale influence on drought and fire activity, effectively setting the "background state" upon which other climatic oscillations modulate interannual and interdecadal fire activity and drought [54]. To further evaluate this hypothesis, a fire occurrence dataset long enough to

capture at least one full AMO cycle would be needed, particularly including a negative AMO phase. It would be instructive to note how the trend, as isolated by Comp0, behaves during a negative AMO phase and whether the large variance attributed to Comp0 persists as well. However, Comp0 may also integrate the warming and drying trend due to climate change that has persisted across the Southwestern US over the past several decades [55], therefore driving greater evaporative demand and fuel drying. We did not explicitly quantify the impact of climate change on AFSI but can reasonably assume that it is a factor.

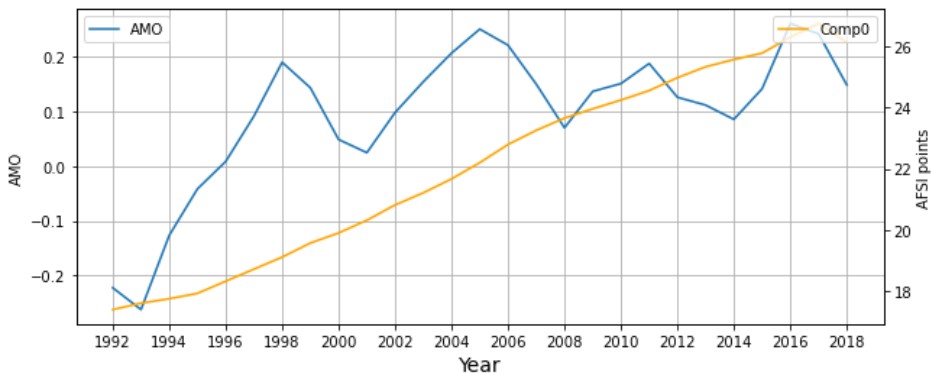

**Figure 7.** Yearly Atlantic Multidecadal Oscillation (AMO) Index (SG filter, window length $l = 61$ months and polynomial order $p = 3$) vs. Comp0 time series for the 27-year study period, 1992–2018.

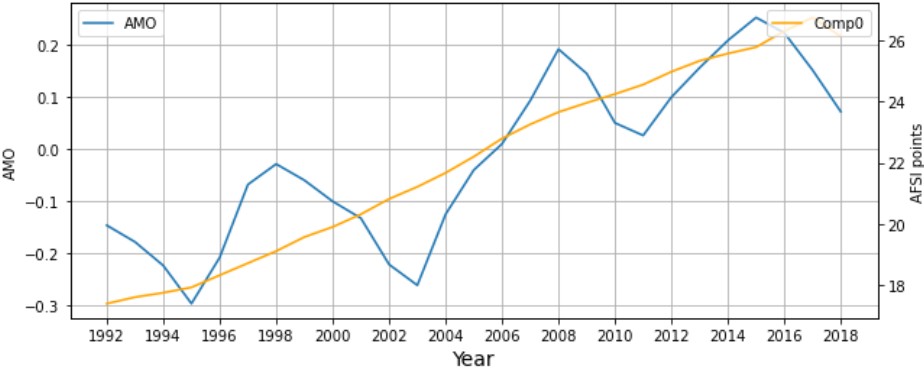

**Figure 8.** Same as Figure 7, but AMO and Comp0 are compared at a 10-year lag, where AMO precedes Comp0. AMO values from the time period 1982–2008 are thus compared to Comp0 values from 1992 to 2018.

4.1.2. Component 1–2

Comp12 exhibited clear oscillatory behavior with a periodicity of 14–15 years with values ranging from $max = 7$ to $min = −7$ and oscillations roughly centered around the $y$-axis (Figure 6). Comp12 is thus an oscillatory component, supporting the SSA decomposition as having effectively isolated the oscillatory components from the secular trend. Comp12 represented 4.91% of the overall time series variance but 55.48% of the non-trend variance (Table 1), indicating the climate signal isolated here is one that imparted significant influence on the SCGA fire regime.

Comp12 is best related to PDO, which exhibited an irregular periodicity within the study period (Figure 9). From 1992 to 1998, a weak positive phase was observed, switching to a prolonged negative phase from 1998 to 2014, though interrupted briefly by a neutral phase in the early 2000s, and finally entering a brief warm positive phase from 2014 to 2017. Comp12 and PDO were closely aligned throughout the first 12 years from 1992 to 2004, though the linear relationship weakened from 2004 to 2018. Accordingly, Comp12 and PDO shared a weak correlation $r = 0.206$ ($p > 0.05$) throughout the study period.

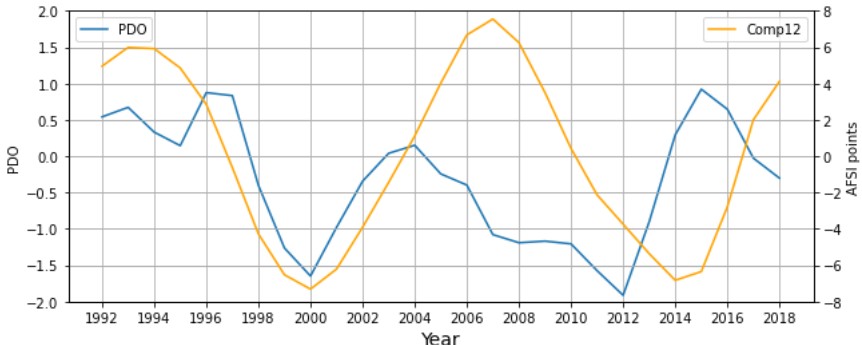

**Figure 9.** Yearly Pacific Decadal Oscillation (PDO; SG-filtered, window length $l = 61$ months and polynomial order $p = 3$ ) vs. Comp12 time series for the 27-year study period, 1992–2018.

However, recalling that positive (negative) phases of PDO tend to indicate anomalously cooler, wetter (warmer, drier) conditions across the SCGA, we would expect a negative correlation between Comp12 and PDO, where positive PDO values are related to reduced regional fire activity and vice versa. Specifically, PDO and AFSI oscillations should be lagged at approximately 1/2 wavelength, roughly in anti-phase. This physical intuition would suggest that a multi-year lag between PDO and Comp12 more accurately represents the fire–climate relationship here.

Figure 10 displays Comp12 and PDO at a 5-year lag, where PDO values from 1987 to 2013 are overlayed against Comp12 from 1992 to 2018. The plot can be separated between a period of stronger and weaker PC–predictor relationships: first, from 1992 to 2005 when the two parameters remained closely in anti-phase, and second, the period of 2006–2018, in which the correlation was weaker because the respective phases were offset by about 1/4 wavelength. Within this second period, PDO oscillations were more irregular but retained a roughly decadal periodicity, tending toward a prolonged negative phase. Though the correlation between these parameters over the full 27-year period was weak at $r = -0.099$ ($p > 0.05$), that correlation was strengthened significantly when considering only the 15 years from 1992 to 2007 with $r = -0.557$ ($p < 0.05$).

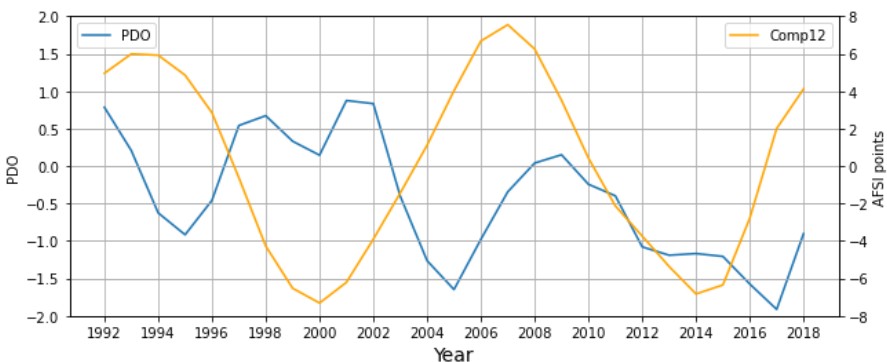

**Figure 10.** Same as Figure 9, but PDO is lagged at 5 years, preceding Comp12 time series. PDO values from the time period 1987–2013 are thus compared to Comp12 values from 1992 to 2018.

Similar fire–climate lags have been established previously in the Western US—in their examination of PDO's influence on fire occurrence in the Pacific Northwest, Hessl et al. [48] noted that the PDO phase precedes the percentage of fire-scarred trees by 5 years, whereas Miller et al. [49] noted a lag of 5–7 years between PDO and fire occurrence in the Sierra Nevada and Southern Cascades. Such a multi-year lag was hypothesized to represent a long-term influence of multidecadal winter moisture patterns on fine fuel condition and abundance via variations in foliar moisture and plant productivity. Our observations support the notion that PDO may primarily modulate the SCGA fire regime through these same physical mechanisms via longer-term moisture anomalies.

The periods of stronger and weaker PC–predictor relationships here coincided with contingent oscillation phases known to modulate PDO's influence on Western US climate. Hidalgo [51] noted that during neutral AMO phases, PDO tends to become more dominant through teleconnections with winter precipitation. Our research found the strongest Comp12–PDO relationships as AMO entered a multi-year neutral phase through much of the 1990s, whereas the weakest relationships occurred as a prolonged AMO-positive phase strengthened through the remainder of the study period. The subsequent period of weakened fire–climate relationships could also be related to the shorter, more irregular PDO cycles that have prevailed since about 1998 [56]. Thus, the tendency for regular PDO oscillations and their attendant multi-year anomalous wet/dry periods to promote fine fuel growth and drying may have been reduced. Additionally, the PDO teleconnections may change with time, possibly modifying the length of the teleconnection lag between PDO phases and attendant SCGA moisture anomalies. Finally, the PCs themselves may be forced differently over time by other oscillations (e.g., AMO, ENSO), another observation made by Hidalgo [51] and elaborated later in the Section 5 herein.

### 4.1.3. Component 3–4

Comp34 exhibited clear oscillatory behavior with a periodicity of 5–6 years, oscillating about the $y$-axis, but with an increasing magnitude over time (Figure 6). Comp34 values ranged from $min = -2$ to $max = 2$ AFSI points during the first oscillation but by $-6$ to 6 points over the latter half of the study period. Comp34 represented 2.39% of the total time series variance and 27.01% of the non-trend variance (Table 1), again representing a climatic component with significant influence on the SCGA fire regime.

Comp34 exhibited the strongest correlation with our annual EDDI threshold exceedance counts (EDDI herein) parameter used as a proxy for short-term, intra-annual drought conditions. The EDDI time series featured an irregular oscillation with a periodicity of 5–8 years and a magnitude increasing by a factor of ~3, closely aligning with the behavior of Comp34 (Figure 11). The Comp34–EDDI correlation was of moderate strength with $r = 0.512$ ($p < 0.05$) throughout the full study period. As with the Comp12–PDO time series, Comp34–EDDI could also be separated into one half that correlated weakly from 1992 to 2006 and another half with a much stronger correlation from 2006 to 2018 ($r = 0.628$, $p < 0.05$) in terms of both periodicity and magnitude. It is noteworthy that the strongest fire–climate relationships here were observed during the weakest such relationships for Comp12–PDO and that the transition point between these time periods also occurred around 2005–2006.

The Comp34–EDDI relationship seemed to be modulated by AMO, PDO, and ENSO collectively. The initial period of weaker Comp34–EDDI correlation was largely characterized by negative/neutral AMO, positive PDO, and mostly positive ENSO; this contingent of oscillation phases generally favored cooler, wetter conditions across the SCGA. During the latter period of stronger Comp34–EDDI correlation, however, AMO maintained a prolonged positive phase, whereas both PDO and ENSO tended toward mostly negative phases. This phase contingency generally favored anomalously warm, dry conditions across the SCGA, and it is evident in the Comp34 time series that higher AFSI values were indeed found in this period (aside from 2010, a 1-year period of positive ONI). A dampening of the Comp34–EDDI oscillation is observed by 2016, as both PDO and ENSO entered a positive phase.

Although our EDDI parameter does not explicitly quantify SAW occurrence and is only weakly correlated to the yearly SAW count time series calculated in Rolinski et al. [15], ($r = 0.293$, $p > 0.05$), it is noteworthy that both the SAW time series and our Comp34–EDDI parameter exhibited similar responses to contingent AMO, PDO, and ENSO phases. The authors noted a period of reduced SAW frequency through 2005, whereas a distinct period of elevated SAW frequency was found from 2006 to 2014. Statistically significant relationships were identified with AMO, PDO, and ENSO. Therefore, our Comp34 appeared to effectively integrate the influence of short-term moisture anomalies due in part to SAWs

while also representing the influence of AMO, PDO, and ENSO on SCGA fire activity. We hypothesize that only a weak Comp34–SAW correlation existed because SAW events often occur throughout the winter months when fuels may be sufficiently moistened to preclude widespread fire activity [15], whereas our EDDI parameter more directly reflects fuel flammability with or without the explicit presence of SAWs.

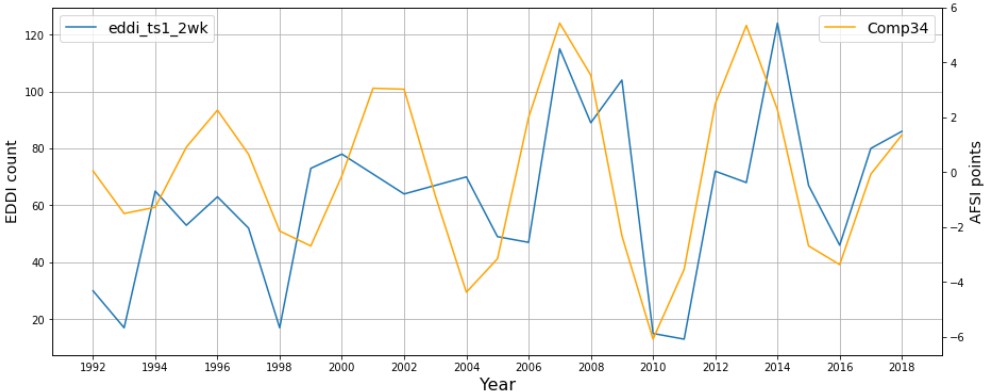

**Figure 11.** Yearly Evaporative Demand Drought Index (EDDI) threshold exceedance counts (# of times per year $EDDI_{2-week} \geq 1$) vs. Comp34 time series for the period 1992–2018.

### 4.1.4. Component 5–6

The final component considered in this analysis, Comp56, is also an oscillatory component with a periodicity of 2–3 years, varying from $min = -6$ to $max = 5$ AFSI points (Figure 6). However, a substantial loss of interannual variability was evident over roughly the second half of the study period, beginning in the mid-2000s, with values ranging from only –1 to 1 points. Comp56 was responsible for 0.98% and 11.07% of the total and non-trend variance, respectively, and was the smallest component of the decomposition (Table 1). Comp56 thus isolated a component of the SCGA fire–climate system that modulated fire activity initially but underwent a significant transition in the mid-2000s by which its mode of variability was effectively overridden by another.

Comp56 correlated most closely with ENSO (Figure 12). Over the full study period, a moderate negative correlation was identified ($r = -0.439$, $p < 0.05$), whereas the relationship strengthened when considering only the first 15 years from 1992 to 2007 ($r = -0.633$, $p < 0.05$). ENSO periodicity here was variable, ranging from 2 to 6 years, with irregular period and magnitude throughout. Although ENSO exhibited greater interannual variability from 1992 to the early 2000s, that regular oscillatory behavior was interrupted by a multi-year period of reduced interannual variability in the early to mid-2000s. Periods of reduced interannual ENSO variability have been linked to reduced fire activity across the Western US due to a lack of anomalous wet/dry periods and subsequent fine vegetation growth, drying, and fuel accumulation [7,57]. This pattern is evident here in the Comp56–ENSO relationship from the early to mid-2000s. Yet, as ENSO returned to a pattern of increased positive and negative phases from 2006 to 2018, no coincident increase in Comp56 variability was observed.

Comp56 behaved similarly to Comp34 in that its strongest PC–predictor correlation occurred during the first half of the study period when AMO remained mostly negative or neutral, whereas ENSO and PDO remained mostly positive. We again identified a significant transition in the early 2000s whereby the dampened and irregular behavior of Comp56 coincided with extended positive AMO and mostly negative ENSO and PDO. Kitzberger et al. [7] reported a statistically significant relationship for synchronous (i.e., widespread) Sierra Nevada fire activity during negative AMO, positive PDO, and positive ENSO. This could be attributed to the observed decadal modulation of AMO on ENSO whereby negative AMO phases lead to stronger ENSO events [58], yielding closer fire–climate relationships. The non-linear modulation of AMO on ENSO was not explicitly captured by our PC–predictor relationships, yet it is intriguing that SSA seemingly captured

the complex breakdown of these fire–climate relationships. Additionally, our research supports the notion that ENSO played a relatively minor role on SCGA fire activity over the 27-year study period, evidenced by the low AFSI variance of Comp56 and the significantly dampened oscillation beginning in the mid-2000s. This aligns with analyses of drought–oscillation relationships in which ENSO exhibited a weaker influence on drought frequency over the Western US than PDO and AMO [8,51].

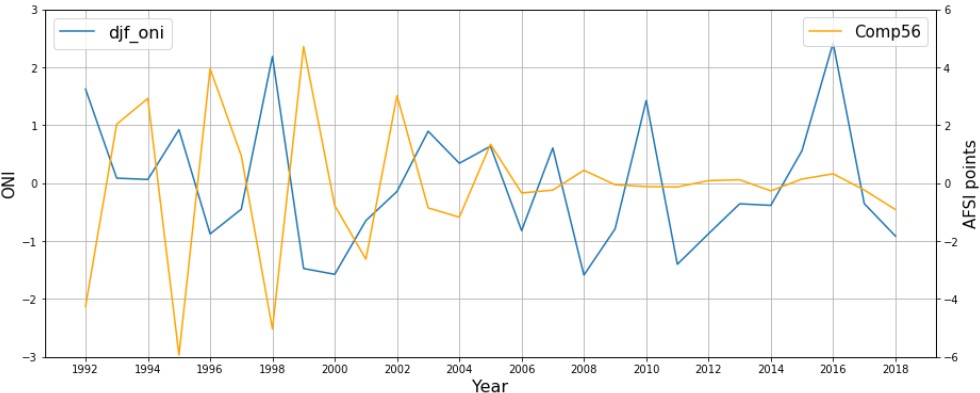

**Figure 12.** Yearly December–February mean Oceanic Niño Index (ONI) vs. Comp56 time series for the period 1992–2018.

## 5. Discussion and Conclusions

Our research employed two important methodological advances in delineating the macroscale fire–climate relationships of Southern and Central California. First, we developed an annual index of fire activity that aimed to dilute the complex interference of anthropogenic influence from within the fire occurrence record by maximizing the influence of climatic metrics in our quantification of fire activity. One of the key benefits of our AFSI methodology lies in its simple calculation and interpretation for fire managers. Each of our index inputs (ignitions, acres burned, ERC magnitude, fire season length) are established within the fire science and fire management communities, and the AFSI output is a normalized numerical value that lends itself to easy historical comparisons. An additional strength of this model structure is its adaptability, where the AFSI can be represented by any set of metrics that quantifies both observed and potential fire activity according to the specific goals of the user. Therefore, future research could investigate whether different fire activity metrics may better account for the anthropogenic manipulation of the fire regime and whether weighting each variable would better represent fire season severity.

As traditional metrics of fire activity decouple from their atmospheric drivers, our research builds upon the understanding that conventional measures of fire activity must be supplemented by indices that more fully encapsulate the variety of ways that wildfires can disrupt society, particularly as the WUI continually expands. One example comes from Beverly et al. [53], who utilized ecological and socioeconomic measures of wildfire activity, including wildfire evacuation frequency and fire suppression expenditure data, to deduce relationships with AMO. The authors found a significant connection between AMO index values and these indicators of fire activity. This method emphasizes the societal impact of wildfires, thereby providing a direct link between research results and their utility in informing fire management and policy decisions. Such metrics could be exploited within the AFSI framework to deduce relationships with climate patterns.

The second methodological improvement tested in this research was the application of SSA to identify the individual influences of climatic predictors on our fire activity index. SSA's model-free, non-parametric technique reconciles the oscillatory behavior of climatic modes and the complex, non-linear fire–climate relationships that may be incompletely captured by other modeling techniques. The results of the PC–predictor relationships suggest that we can indeed isolate the signal of individual climatic predictors on the AFSI. The top 7 PCs (Comp0–6) responsible for the greatest variance in the reconstructed time

series were each correlated with respective climate predictors and generally exhibited statistically significant correlations of moderate strength $|r| > 0.4$. Each of the three oscillatory components (Comp12, 34, and 56) could be separated into distinct periods: about one half of the time series where PC–predictor correlations were moderate–strong and another half where the correlations were generally weak. For each of these scenarios, we were able to suggest how respective phases and phase transitions of AMO, PDO, and/or ENSO may have strengthened or weakened these correlations.

One of the complicating factors in this methodology may have been related to the construction of the PCs themselves and the assumptions contained within this methodology. Illustrating this obstacle is Hidalgo [51], who employed a variant of PCA to identify the dominant modes of variability of the Palmer Drought Severity Index (PDSI) over the Western US. The author found that each of the resulting PCs, which are by definition, orthogonal, were correlated to AMO and PDO. However, the author determined that the forcings of each oscillation can influence the other's state and are therefore intercorrelated to some degree. Thus, the relative influence of one oscillation from another on PDSI cannot be fully untangled. Therefore, the "identification of PCs is achieved at the expense of losing some of the characteristics (i.e., intercorrelation) known to exist between the main teleconnection signals" [51]. Over time, the PCs may be forced differently by each climatic oscillation, potentially resulting in stronger or weaker PC–predictor relationships. Such a case in our work can be identified in the supposed interaction between AMO and ENSO, with influences on the Comp56–ENSO relationship. During the first half of the study period, when AMO remained in a negative phase, the strongest Comp56–ENSO relationship was observed. Because negative AMO phases have been shown to lead to stronger ENSO events [58], we hypothesize that the subsequent climate patterns induced by such ENSO events during this time would tend toward more pronounced wet–dry cycles in our study region. Kitzberger et al. [57] showed that these pronounced ENSO effects play a significant role in enhanced regional fire frequencies. Therefore, we can surmise that AMO's implicit influence on ENSO modified the Comp56–ENSO relationship, which we cannot currently explicitly identify. However, we suggest that the signal of fire–climate relationships herein remains promising. SSA's ability to resolve complex interactions between the dominant modes of fire–climate variability is evident in our study of the SCGA.

Finally, SSA also helped to elucidate significant shifts in the dominant modes of SCGA fire–climate variability in the mid-2000s that align with previous findings, e.g., [50,59]. This is a significant strength of the technique—the data-adaptive structure allows for the identification of time series that exhibit dominant structures that differ along the signal [45]. It is intriguing that such a shift in fire–climate modes of variability were detected in each of our oscillatory components and that these observations could be leveraged to better resolve the complex interactions of individual climatic oscillations. We believe that with continued research, this method can be further developed into a predictive model of annual fire activity for any macroscale fire regime given inputs of the region's dominant climatic features. Such a model would have the most utility in the months leading up to each fire season to proactively determine resource requirements per geographic area, thereby optimizing resource efficiency nationwide.

**Author Contributions:** Conceptualization, T.R. and B.B.; methodology, T.R., B.B., E.H. and E.K.; software, B.B. and D.B.; formal analysis, B.B.; investigation, B.B.; data curation, B.B.; writing—original draft preparation, B.B.; writing—review and editing, B.B., T.R., E.H. and E.K.; supervision, T.R., E.H., E.K. and D.B. All authors have read and agreed to the published version of the manuscript.

**Funding:** This research received no external funding.

**Data Availability Statement:** Data are available upon request from the corresponding author.

**Conflicts of Interest:** The authors declare no conflict of interest.

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
