# Peer review of "Refining Fire–Climate Relationship Methodologies: Southern California"

_fire, doi:10.3390/fire6080302_

Round 1
Reviewer 1 Report
The article mention that 95% fire is human made, but not clear what kind of it clearly written and the rest 5% is also not clear what cause it. There should be clearly written also about the trend fires often happened with or without El_nino.
Author Response
Thank you for your comment. We assume you are referring to the first sentence in the Introduction section of the manuscript. “In Southern California, where it is estimated that ~95% of all wildfire ignitions are human caused [1], trends in fire activity are often decoupled from the climate and weather patterns that shape the regional fire regime.” We are not sure what is unclear. We are simply stating that approximately 95% of wildfires in Southern California are human caused and that the other 5% can be attributed to natural causes. Are you asking for a more specific breakdown of wildfire cause?
Reviewer 2 Report
The manuscript is a good one. The methodology is clear. Results are reliable. Objectives and conclusions are relevant. The reference section is also rich. It should be definitely accepted.
Author Response
Thank you for your review of our manuscript.
Reviewer 3 Report
The authors survey the refinement of fire-climate relationship methodologies in southern California. The study is rigorous and the article is well structured; however, I have a few comments that would help improve the quality of the article.
1. It would be good if the authors could elaborate on the measurable relationships between wildfire activity and regional fire-climate relationships. Justify how the four fire activity metrics have been chosen
2. How the authors delineate macroscale fire-climate relationships in southern and central California.
3.Why have the authors not considered giving different weights to ignitions, hectares burned, ERC magnitude, fire season length, according to their potential influence in quantifying fire activity?
4.Have the authors considered studying different metrics of fire activity to see if they can better reflect anthropogenic manipulation of the fire regime?
5. How might climate oscillation lead to stronger or weaker PC-predictor relationships that have influenced signal outcomes and hence fire-climate relationships?
Author Response
- It would be good if the authors could elaborate on the measurable relationships between wildfire activity and regional fire-climate relationships. Justify how the four fire activity metrics have been chosen.
We thank you for your review of our manuscript. To justify how the four fire activity metrics were selected, we have stated in the manuscript the following:
“Our AFSI is comprised of annual values of each of the four components, two derived from the FPA FOD (acres burned, ignition counts), and two derived from the ERC database (ERC magnitude, fire season length). These four components were preliminarily selected based on their perceived ability to quantify different but equally important modes of annual wildfire severity, related to observed fire activity (which contain an element of anthropogenically-derived stochasticity) and potential fire activity (which are related only to atmospheric conditions and the attendant impacts on fuel flammability). Each of the four metrics are established throughout the fire science community and can be easily attained and interpreted.
An exploratory analysis was conducted on each of the components’ empirical distributions to identify their most appropriate quantification, keeping in mind that we aim to comprehensively represent the different modes of annual fire activity and their potential impact on fire suppression efforts. The first component, acres burned, was calculated as the annual number of fires exceeding the climatological (1992-2018) 97th percentile fire size. This focuses on the annual fire activity contribution by very large fires which can require a disproportionate suppression resource response compared to smaller fires. The second component, annual ignition count, is left unaltered because it directly represents the potential for many concurrent fires to strain fire suppression resources. Third, we determined that the ERC magnitude component is best represented by the yearly 90th percentile ERC value. Daily ERC values, which range from about 10-95 throughout the study period, display a right-skewed distribution, indicating that high daily ERC values are more frequent. The 90th percentile value is less prone to sampling variation than a lower percentile, such as the median, because of this right skew.
The calculation of the fourth AFSI component, fire season length, was necessarily more subjective due to the complex nature of ERC time series and the fuel conditions they represent. Highly variable winter rainfall interspersed with frequent SAW events across Southern California commonly result in fuel moisture values that fluctuate greatly from day-to-day in the cool season. We chose to emphasize the assumed strain on suppression resources of many consecutive days with elevated fire potential. The daily ERC time series was smoothed with a Savitzky-Golay (SG, [32]) filter of window length days and polynomial order . This filter is advantageous in its ability to preserve narrow time series peaks better than many other smoothing filters with Gaussian convolution [33]. We then calculated the annual number of consecutive days where , an empirical threshold under which we determined significant fire activity was unlikely. However, it is common for multi-day periods to exceed this empirical threshold outside of the primary fire season; therefore, these sub-periods were included in the respective year’s fire season count if they remained within 15 days of the beginning or end of the primary fire season. Fire season tallies were cut off on December 31st of each year.”
We believe these paragraphs thoroughly and succinctly justify our choice of fire activity metrics. For each of the four respective metrics, we state how the particular characteristics of observed or potential fire activity (size, fuel flammability, suppression requirements, etc.) can represent scenarios which may influence subsequent fire activity-climate relationships.
- How the authors delineate macroscale fire-climate relationships in southern and central California.
Great question. We use the term “macroscale fire-climate relationships” here to refer to such relationships across Geographic Areas (GAs; as defined by the National Wildfire Coordinating Group). To clarify this, we have added the following to our manuscript at the end of section 2, “Study Area”:
The term “macroscale” is used here to differentiate from the small scale, on-the-ground influences of factors including heat and moisture fluxes on fire activity which are not the focus of this research. Our research focuses on macroscale effects of climate patterns on fire activity across broad fire regimes, relationships which are much more readily established at large geographic scales.
- Why have the authors not considered giving different weights to ignitions, hectares burned, ERC magnitude, fire season length, according to their potential influence in quantifying fire activity?
We did consider weighting the variables but that would entail much more analysis with likely little difference in the overall results. And considering that this study is a first look and “exploratory”, we felt this level of detail was unnecessary at this time. We did modify the text in the Discussion and Conclusion section to say:
“Therefore, future research could investigate whether different fire activity metrics may better account for the anthropogenic manipulation of the fire regime, and if weighting each variable would better represent fire season severity.”
- Have the authors considered studying different metrics of fire activity to see if they can better reflect anthropogenic manipulation of the fire regime?
We did consider studying different metrics of fire activity to account for anthropogenic manipulation of the fire regime. As stated in the manuscript, we wanted to represent both the observed and potential impacts of fire activity using metrics that are “established within the fire science and fire management communities” and are simply “calculate[ed] and interpret[ed] by fire managers”. Fire-climate research frequently employs statistics of acres burned and number of ignitions, and we felt it important to develop a methodology with roots in already-established methodologies. Therefore, these first two metrics were included. The final two metrics related to the Energy Release Component, which serve to quantify potential fire activity, are indicators of fuel flammability. ERC has been shown to relate strongly to significant fire activity, stated in the manuscript: “ERC is closely related to large fire occurrence and extreme fire behavior [2]”. These metrics thus relate more closely to weather and subsequent fuels conditions, therefore reducing the influence of anthropogenic activity on the fire occurrence record.
Finally, the purpose of our research is exploratory, to introduce a useful methodology while remaining concise and focused. We stress that it is not just the results of our research, but the framework of the methodology used to calculate our Annual Fire Severity Index (AFSI), which we believe holds significant utility. We state in the Discussion/Conclusion section:
“An additional strength of this model structure is its adaptability, where the AFSI can be represented by any set of metrics that quantify both observed and potential fire activity according to the specific goals of the user. Therefore, future research could investigate whether different fire activity metrics may better account for the anthropogenic manipulation of the fire regime, and if weighting each variable would better represent fire season severity.
As traditional metrics of fire activity decouple from their atmospheric drivers, our research builds upon the understanding that conventional measures of fire activity must be supplemented by indices that more fully encapsulate the variety of ways wildfires can disrupt society, particularly as WUI continually expand. One example comes from Beverly et al. [53], which utilized ecological and socioeconomic measures of wildfire activity including wildfire evacuation frequency and fire suppression expenditure data to deduce relationships with the AMO. The authors found a significant connection between AMO index values and these indicators of fire activity. This method emphasizes the societal impact of wildfires, thereby providing a direct link between research results and their utility in informing fire management and policy decisions. Such metrics could be exploited within the AFSI framework to deduce relationships with climate patterns.”
- How might climate oscillation lead to stronger or weaker PC-predictor relationships that have influenced signal outcomes and hence fire-climate relationships?
Good question. While we do not explicitly identify the complex interplay between climate oscillations and the subsequent impacts on fire activity, as this remains outside the scope of our research, we do address this issue in the Discussion. Drawing from Hidalgo [51], we state:
“The author found that each of the resulting PCs, which are by definition orthogonal, were correlated to AMO and PDO. However, the author determines that the forcings of each oscillation can influence the other’s state and are therefore intercorrelated to some degree. Thus, the relative influence of one oscillation from another on PDSI cannot be fully untangled. Therefore, the “identification of PCs is achieved at the expense of losing some of the characteristics (i.e. intercorrelation) known to exist between the main teleconnection signals” [51] . Over time, the PCs may be forced differently by each climatic oscillation, potentially resulting in stronger or weaker PC-predictor relationships. It is likely that similar complications have impacted our findings…”
To more specifically draw from our work, examining the interplay between AMO and ENSO on fire activity in Southern California, we have added the following to our Discussion:
“Such a case in our work can be identified in the supposed interaction between AMO and ENSO with influences on the Comp56-ENSO relationship. During the first half of the study period when AMO remained in a negative phase, the strongest Comp56-ENSO relationship was observed. Because negative AMO phases have been shown to lead to stronger ENSO events, we hypothesize that the subsequent climate patterns induced by such ENSO events during this time would tend toward more pronounced wet-dry cycles in our study region. Kitzberger et al (2001) has shown that these pronounced ENSO effects play a significant role in enhanced regional fire frequencies. Therefore, we can surmise that AMO’s implicit influence on ENSO has modified the Comp56-ENSO relationship, for which we cannot currently explicitly identify.”
Reviewer 4 Report
The subject of the article, the methods used and the text of the article itself is very interesting. A review of the literature is sufficient. Descriptions of the results obtained, whether the conclusions are clear.
However, when I read the article for the first time, I had a problem determining what exactly is given, what is counted. Of course, everything in the work is described, but it may not be clear enough for the reader (the reader does not have to be a statistician). Actually, I'm not a supporter of block diagram, but maybe in this case, when the approach is innovative, it would make sense to present in the form of block diagram how to proceed in this work. I leave the decision in this matter to the authors, but I would like to ask for a clearer, short text regarding the scope of this work.
It also seems to me reasonable to include in the Methods chapter a short description of Singular Spectrum Analysis (SSA).
In many places, the authors write that the variables are correlated - more or less. I would suggest using a more specific approach: which of the correlation coefficients was calculated (Pearson, Kendall or Spearman) and whether this coefficient is statistically significant at the assumed (a = 5%?) significance level or not - please specify it in the Methods chapter.
Author Response
The subject of the article, the methods used and the text of the article itself is very interesting. A review of the literature is sufficient. Descriptions of the results obtained, whether the conclusions are clear.
Thank you for your review.
However, when I read the article for the first time, I had a problem determining what exactly is given, what is counted. Of course, everything in the work is described, but it may not be clear enough for the reader (the reader does not have to be a statistician). Actually, I'm not a supporter of block diagram, but maybe in this case, when the approach is innovative, it would make sense to present in the form of block diagram how to proceed in this work. I leave the decision in this matter to the authors, but I would like to ask for a clearer, short text regarding the scope of this work.
We have created a block diagram to clarify the workflow of the AFSI development, from initial fire occurrence and ERC datasets to the final AFSI time series.
Regarding your final statement, “I would like to ask for a clearer, short text regarding the scope of this work”, are you recommending we clarify and shorten the manuscript as a whole, or rather are you suggesting the addition of a block diagram to understand the construction of our AFSI is sufficient? If you are recommending we clarify and shorten the manuscript, can you point to any specific sections that need to be clarified and/or shortened?
It also seems to me reasonable to include in the Methods chapter a short description of Singular Spectrum Analysis (SSA).
We do provide about two paragraphs that describe Singular Spectrum Analysis in section 3.2.1. We believe the depth of our discussion is appropriate and provides a like level of detail regarding SSA as do other manuscripts with a similar focus. We further provide the reader with multiple references that describe the technical details of SSA. Our manuscript states:
“To decompose the AFSI time series we employed Singular Spectrum Analysis (SSA; [40–44]) a technique which has been used extensively on geophysical and climate data. SSA decomposes a time series into its underlying components (trend, oscillatory modes, noise), and is especially useful for decomposition of short, noisy time series. As a non-parametric spectral estimation method, SSA makes no assumptions about the parametric form of the time series. SSA uses data-adaptive basis functions and therefore can separate potentially irregular oscillatory behaviors with varying periodicities [45]. This is advantageous because the atmospheric features that control SCGA weather and climate vary irregularly and may not be clearly identified by decomposition methods that assume sinusoidal behavior. SSA is essentially Principal Component Analysis (PCA) on a time series, identifying the primary modes of temporal variability from within the data. The main result is the set of orthogonal reconstructed components (RCs), herein referred to as principal components (PCs) which together sum to the original time series.
SSA is also practical and easy to employ because it requires only one variable, the “window length”, or the number of consecutive time series observations, n, upon which a delay window is applied, and determines the longest periodicity captured by SSA. Given a window length of L, SSA will produce L principal components. Larger values of L provide more refined decomposition into elementary components and therefore better separability [46]. To sufficiently extract the underlying trend of the AFSI, a window length of was chosen. Therefore, the window length consisted of about 44% of the overall study period length. The SSA code implemented herein is adapted from a Python script by D’Arcy [47].”
Is there a specific aspect of SSA that the reviewer would like our manuscript to expand on within this description from our Methods section?
In many places, the authors write that the variables are correlated - more or less. I would suggest using a more specific approach: which of the correlation coefficients was calculated (Pearson, Kendall or Spearman) and whether this coefficient is statistically significant at the assumed (a = 5%?) significance level or not - please specify it in the Methods chapter.
We do state that the Pearson correlation coefficient was used to establish the strength of the calculated fire-climate relationships at the end of section 3.2.1:
“Times series of each predictor and PC are first plotted to observe patterns within the data, and the relationships are then quantified using the Pearson correlation coefficient (.” Additionally, in each subsection of the Results for respective PC-predictor pairs, we provide both the Pearson correlation coefficient and the p-value. One example is found in the second paragraph of section 4.1.1. when we state: “Comp0 was identified here as strongly correlated to AMO ()”.
Reviewer 5 Report
The authors examine an interesting topic (Refining fire-climate relationship methodologies: Southern Californi). The introduction section is needs improvement. In additions, more results should be included in the abstract section. The discussion and conclusion section should be separately according the journal format, more importantly the author should recommend the implications of the study for policy makers; otherwise it seems like hypothetical study without field based utility. However, the data in the manuscript deserve publication and thus I give some advice with which the authors could add in the manuscript that could be published for example in this journal.

Author Response
“The authors examine an interesting topic (Refining fire-climate relationship methodologies: Southern California). The introduction section is needs improvement.”
- Thank you for your review of our manuscript. In the pdf attached to your review, we noted that your only suggestion for improving the Introduction was to swap the first and second paragraphs so that the Introduction would begin with the description of the role that weather and climate play on fire activity generally, and in our study region. You recommend the second paragraph should then describe the influence of anthropogenic activity on fire activity in Southern California. While we understand your reasoning for this suggestion, namely to open the manuscript with an emphasis on how fire is influenced by weather and climate, we believe the original Introduction configuration is most appropriate. Our research was motivated by the difficulty in establishing fire-climate relationships in Southern California because of the extensive stochastic element imparted on fire activity by anthropogenic influence. While our research discusses weather and climate influences on fire activity extensively, we do this with a primary goal in mind--to minimize the influence of anthropogenic influence on the fire occurrence record, thus clarifying fire-climate relationships. Therefore, we believe it is most appropriate to open the manuscript by emphasizing to the reader the motivating conflict that inspired our research.
“In additions, more results should be included in the abstract section.”
- Thank you for your suggestion. We have added to the abstract a brief description of additional results that identified a transition between weaker and stronger fire-climate relationships in the mid-2000s. Our description of the results and conclusions in the abstract now reads:
“The results indicate that SSA effectively isolates the individual influence of each predictor on AFSI quantified by generally moderate fire-climate correlations, , over the full study period, and over select 13-15 year periods. A transition between weaker and stronger fire-climate relationships for each of the oscillatory PC-predictor pairs was centered around the mid-2000s, suggesting a significant shift in fire-climate variability at this time. Our approach of aggregating and decomposing a fire activity index yields a straightforward methodology to identify the individual influence of climatic predictors on macroscale fire activity even in fire regimes heavily modified by anthropogenic influence.”
“The discussion and conclusion section should be separately according the journal format.”
- Per the journal website, there are no strict formatting requirements that require the discussion and conclusion to be separate. At the journal website (https://www.mdpi.com/journal/fire/instructions), it is stated “We do not have strict formatting requirements, but all manuscripts must contain the required sections: Author Information, Abstract, Keywords, Introduction, Materials & Methods, Results, Conclusions, Figures and Tables with Captions, Funding Information, Author Contributions, Conflict of Interest and other Ethics Statements.” We do include a Conclusion within the Discussion/Conclusion section. The website also states “Conclusions: This section is not mandatory but can be added to the manuscript if the discussion is unusually long or complex.” Therefore, we believe it is appropriate to format the Discussion/Conclusion section as we have in our manuscript.
“…more importantly the author should recommend the implications of the study for policy makers; otherwise it seems like hypothetical study without field based utility.”
- Thank you for this helpful suggestion. We have added the following statement to the final paragraph of our Discussion/Conclusion to more directly address the implications of this study for policy makers.
“We believe that with continued research, this method can be further developed into a predictive model of annual fire activity for any macroscale fire regime given inputs of the region’s dominant climatic features. Such a model would have the most utility in the months leading up to each fire season to proactively determine resource requirements per Geographic Area, thereby optimizing resource efficiency nation-wide.
“However, the data in the manuscript deserve publication and thus I give some advice with which the authors could add in the manuscript that could be published for example in this journal.”
Thank you very much for your review.
Additional comments from attached pdf:
- Include in Methods section the software used to create box-and-whisker plot of SCGA monthly fire ignitions and ERC magnitude (line 222)
- As we only include one box-and-whisker plot and it is a minor detail to our overall results, we do not believe it is necessary to include this in our Methods.
- As we only include one box-and-whisker plot and it is a minor detail to our overall results, we do not believe it is necessary to include this in our Methods.
- Include discussion of why we chose Singular Spectrum Analysis (SSA), a variant of Principal Component Analysis (PCA), over other ordination techniques like DCA (presumably the reviewer is referring to Detrended Correspondence analysis), etc. (line 320)
- We believe the best approach to describing our methodology is to discuss why we chose the statistical analysis method we did use, not to discuss why we didn’t choose any given analysis method. It would add significant unnecessary descriptions to explain why we didn’t choose the numerous statistical approaches available for any given scenario. We aim to keep our manuscript as concise as possible, and therefore believe that our current discussion of SSA is appropriate.
- We believe the best approach to describing our methodology is to discuss why we chose the statistical analysis method we did use, not to discuss why we didn’t choose any given analysis method. It would add significant unnecessary descriptions to explain why we didn’t choose the numerous statistical approaches available for any given scenario. We aim to keep our manuscript as concise as possible, and therefore believe that our current discussion of SSA is appropriate.
- Include PCA diagram in the results section (line 375)
- We would like to stress that we did not use Principal Component Analysis but rather a variant known as Singular Spectrum Analysis to examine the dominant modes of temporal variability within our Annual Fire Severity Index. The result of the SSA decomposition is the set of temporal Principal Components (PCs). We have provided in our manuscript Figure 5 and Table 1 which collectively provide the relevant statistics that describe the SSA decomposition, including the percent of overall time series variance for which each SSA component is responsible.
- We would like to stress that we did not use Principal Component Analysis but rather a variant known as Singular Spectrum Analysis to examine the dominant modes of temporal variability within our Annual Fire Severity Index. The result of the SSA decomposition is the set of temporal Principal Components (PCs). We have provided in our manuscript Figure 5 and Table 1 which collectively provide the relevant statistics that describe the SSA decomposition, including the percent of overall time series variance for which each SSA component is responsible.
- Cite previous study (line 606)
- We are unsure what the reviewer is referring to here. This comment is attached to a part of the paper where we suggest directions for future research; therefore, there are no previous studies to cite as the research has not been conducted.
- We are unsure what the reviewer is referring to here. This comment is attached to a part of the paper where we suggest directions for future research; therefore, there are no previous studies to cite as the research has not been conducted.
- In Discussion/Conclusion section, compare results with previous studies from other parts of the world.
- As best as the authors can tell, our methodology is novel and has not been employed in other parts of the world. Therefore, we cannot provide explicit comparisons of our results to that of other studies globally. However, we do mention earlier in the Discussion/Conclusion section two examples of how our methods/results compare to other studies. We first suggest that our AFSI methodology could follow the methodology of Beverly et al. [53], research based out of Canada, and incorporate ecological and/or socioeconomic measures of wildfire activity including wildfire evacuation frequency and fire suppression expenditure data. Second, we compare the complications noted in Hidalgo [51] in the construction of Principal Components describing dominant modes of Western US drought variability to that of our research identifying the dominant modes of climate variability on Southern California fire activity. We believe these two examples adequately compare our research to that of similar research in other regions of the world.
Round 2
Reviewer 3 Report
The authors have rigorously answered all the questions raised and the article has improved the article considerably.
I have nothing more to add.